# Investigation on the strategy of business administration level of enterprises in the new economic environment

**Zhuolin Xiao** *

School of Foreign Languages and Economics, Qingyuan Polytechnic, Qingyuan, Guangdong, China

* xiaozhuolin2017@126.com

## Abstract

In the face of the new economic environment, enterprises must continuously enhance their capabilities to achieve long-term development. In the current market scenario, business management relies on economic principles and legal accounting. Considering the current market situation, the article analyzed enterprises system reform and production planning, proposing corresponding countermeasures. Therefore, in order to achieve rapid development, it was necessary to strengthen the management of enterprises. In this paper, the current problems faced by enterprises, solutions and the significance of enterprises needed to improve their management level were explained, and the situation of enterprises was analyzed through the enterprise strategic management model. Comparing with the traditional management model in terms of the complexity of enterprise management processes, efficiency, management level score, and quarterly profit, findings reveal that the management model in the new economic environment has reduced the complexity of the enterprise process by 0.17 points. The management efficiency has increased by 0.15 points, the management score has increased by 14 points, and the quarterly profit of the company has increased by 30,000 yuan. Furthermore, it is elucidated that, in the new economy, enhancing the management level is essential for enabling enterprises to attain long-term development.

## Introduction

Under the new market economic situation, in the increasingly fierce market economic environment, if the company wants to develop and grow steadily in the increasingly competitive market economic environment, it needs to continuously improve its operating strength and further improve its own management system. This is essential to improve the company's business management level. However, in practice, business management in China has caused many problems due to the unsound strategic planning and work flow of the company. In China's e-commerce market, for example, Alibaba, JD.com and Pinduoduo compete fiercely. Enterprises maintain competitive advantages by improving product quality and innovating marketing strategies. Therefore, to do a good job in the business management of enterprises,

**Data Availability Statement:** Uploaded as supplementary information.

**Funding:** This work was supported by the young innovative talents project in Guangdong Province in 2023 "Research on the High-quality Development

Path of Qingyuan Chicken Industry from the Perspective of Industrial Integration" (Project NO. 2023WQNCX249).

**Competing interests:** The authors have declared that no competing interests exist.

that is, not only can rationally arrange production planning, but also seize market opportunities, and lay a solid foundation for the company to carry out management reform and technological innovation.

In the international context, Chinese enterprises are constantly facing new opportunities and challenges. In today's rapid economic growth, enterprises need to speed up the pace of their own business management. This can not only promote the enterprise to keep pace with the times, but also meet the needs of the diversified development of the society, and also promote the enterprise to continuously deepen the reform of the enterprise system and establish a modern development model in the process of internationalization. Manufacturing in China has gained a competitive advantage due to low cost and mass production, but quality management is an important issue. A typical example is Huawei, which builds a good brand through technological innovation and strict quality management.Therefore, enterprises must have a sound business management system, which is of great practical significance to adapt to the development of the times.

Under the background of globalization and the rapid development of information technology, enterprises are faced with a complex and changeable economic environment. Challenges and opportunities: Globalization brings vast markets and competitive pressures, technological advances increase efficiency and innovation, and social responsibility and sustainable development become the keys to long-term success. Enterprises should actively respond by strengthening strategic planning, optimizing management, promoting innovation, and fulfilling social responsibilities to meet the economic needs of the new era. This paper firstly expounds the current problems faced by enterprises, puts forward corresponding countermeasures, and points out the importance of improving management level.It also analyzes the enterprise with its strategic management mode, and points out that in light of the evolving economic landscape, the enterprise must enhance its management level to achieve long-term development.

This paper first introduces the challenges faced by enterprises under the new market economy conditions, and emphasizes the importance of improving management level. Then it discusses the relevant work and scholars' views, and then discusses the management strategy of enterprises in the new economic environment, including the main role of employees, brand value reconstruction and so on. Finally, it puts forward the concrete measures to improve the enterprise management level, and emphasizes the advantages of the new management model. On the whole, this paper systematically expounds the management strategy and practice of enterprises how to achieve long-term development in the new economic environment. The overall structure of the article is shown in Fig 1.

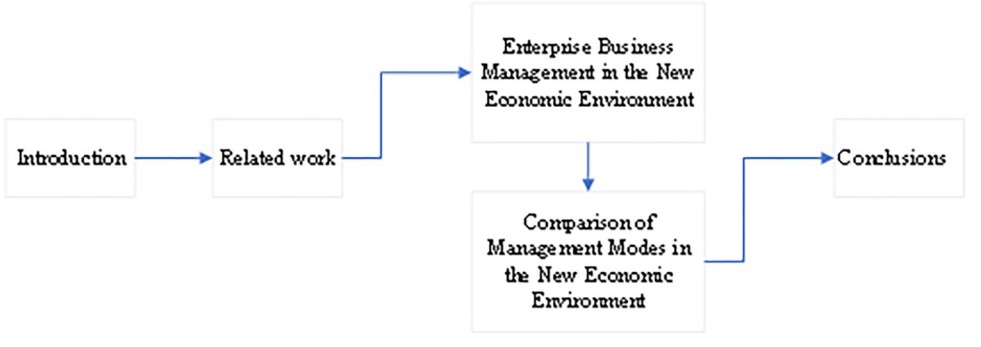

**Fig 1.**

## Related work

In the new economic environment, many enterprises are faced with the management problems of enterprises, and many scholars have discussed the problems of business management of enterprises in the new environment.The main issues of concern include the relationship between corporate innovation and social responsibility, the impact of corporate governance on strategic management accounting disclosure, the expansion of strategic management and corporate risk, and the integration and management process of strategic corporate social responsibility. Under the new normal of the economy, the growth momentum of enterprises is lacking, the internal structure of enterprises is imbalanced, talents are lacking, capital efficiency is reduced, and the phenomenon of ecological damage is more serious.Ying M J investigated the interplay between innovation and corporate social responsibility by delineating their definitions, evolution, and the concept of "responsibility innovation." The study underscored the imperfections in the indicator system and highlighted how addressing issues in corporate innovation and social responsibility could enhance enterprise development [1]. Honggowati S examined the influence of corporate governance on strategic management accounting disclosure, The research results show that the significance value of independent board size is greater than 0.05 (0.745 > 0.05), so H1b is rejected. This may be due to the relatively small number of independent boards, rather than the large number of non-independent directors. In addition, regarding management ownership, the study demonstrated that management ownership has a negative impact on the degree of SMA (H1c is rejected) [2]. In addressing the expanding field of strategic management, Durand R deliberated on its boundaries and implications for enterprises, emphasizing the need for coherent integration and rational deployment within the strategic landscape [3]. Vitolla F builds a dynamic corporate social responsibility (CSR) integration model based on social management philosophy, emphasizes the importance of CSR in strategic management, provides a new way for enterprises to realize their strategic intentions, promote innovation, and enhance the positive impact of economic activities. It has promoted social development [4].

Despite the notable lacuna in research on enterprise management strategies, scholars have shown considerable interest and engaged in discussions on the topic. For instance, Pramanik P proposed a model aimed at identifying optimal business strategies for seasonal sales and sale items, effectively addressing enterprise strategy concerns by leveraging fuzzy and rough event measures to align product prices with customer demand [5]. Furthermore, Hossain M investigated the impact of narrative disclosure features on audit fees and found that they had a positive impact on audit pricing and showed different effects at different stages of the company's transformation. It emphasizes the dynamic nature of the firm and the importance of considering the stage of the firm's life cycle when analyzing the relationship between narrative disclosure characteristics and audit costs [6]. In the pursuit of proficiency in skills and the relevance of information systems to business management in the digital age, Sihite M conducted an extensive review, uncovering a significant correlation between skills and successful business management [7]. Wang H et al. verified the positive impact of digital transformation strategy on organizational performance, and revealed the moderating role of cognitive conflict, which provided in-depth analysis and theoretical support for Chinese enterprises to implement digital transformation, and provided new theoretical perspectives and inspirations for IT/IS field and digital strategy research [8]. Zhang X et al. revealed the importance of IT infrastructure to digital transformation, and emphasized the mediating role of digital transformation strategy and management [9].Through these diverse research endeavors, scholars contribute to a deeper understanding of enterprise management strategies and their dynamic interplay with other organizational components.To bridge the gap between the exploration of skills mastery

and the relevance of information systems to modern business management, it's essential to understand how these elements intersect in the context of the digital age. While Sihite M's analysis sheds light on the correlation between skills and successful management, it prompts further inquiry into how businesses adapt to evolving economic landscapes. Understanding the correlation between skill mastery and effective management can serve as a foundation for addressing the broader challenges and opportunities presented by the digital era. This includes recognizing the significance of strategic management practices in leveraging both human capital and technological innovations to navigate dynamic market environments. By exploring avenues to enhance skill sets and optimize information systems, enterprises can bridge absolute and relative gaps in their management strategies, thereby fostering resilience and adaptability in the face of contemporary economic shifts.

## Enterprise business management in the new economic environment

### Significance of improving the level of business management of enterprises

The new economy is a high-tech economic form driven by the information technology revolution in the context of global economic integration. The operation of the company plays an important role in the development of the enterprise market. Its function is shown in Fig 2. In addition, the business operation of the enterprise must also analyze the daily data, and formulate corresponding plans based on the content of these data to achieve the company's development goals.

Under the new market economy conditions, the development of enterprises has higher and higher requirements for managers. Therefore, enterprises must take the promotion of industrial and commercial administration as an important task [10,11]. How to improve the management level of enterprises has become a concern of major companies. At present, most corporate executives in China are generally less educated, and some are even "outsiders" who lack corresponding management knowledge and skills. Therefore, strengthening the operation and management of enterprises must start with improving the quality and management ability of management personnel. Managers can be promoted through business training. With the development of the times, the development of enterprises requires more and more knowledge of business management. With China's entry into the WTO, the development environment of enterprises has become more and more international, and the competition has become more and more fierce. Under the new economic situation, how to cultivate a group of high-quality management talents is a must for a company to keep pace with the times and adapt to the development trend of the society. In addition, the state also has new requirements for the management of enterprise operators, aiming at cultivating and improving the management level of enterprises to realize the transformation of enterprises. With the development of the economy and the continuous improvement of the market economic system, many companies are faced with huge development opportunities and also a huge challenge. After the establishment of the market economy system, the government has less and less supervision of the market. How to establish a benign market operation mechanism in the market and improve its operating efficiency is an urgent problem to be solved. In the administrative procedures, the concept of business operation is used to standardize the administrative subjects and make greater contributions to the development of the company [12]. It can be seen that strengthening the level of business management of enterprises is also a project of great social significance. Its significance mainly has the following points, as shown in Fig 3.

The modernization requirements of the development are realized.Alibaba is building a digital ecosystem to adapt to changes in market demand and improve the efficiency of enterprise

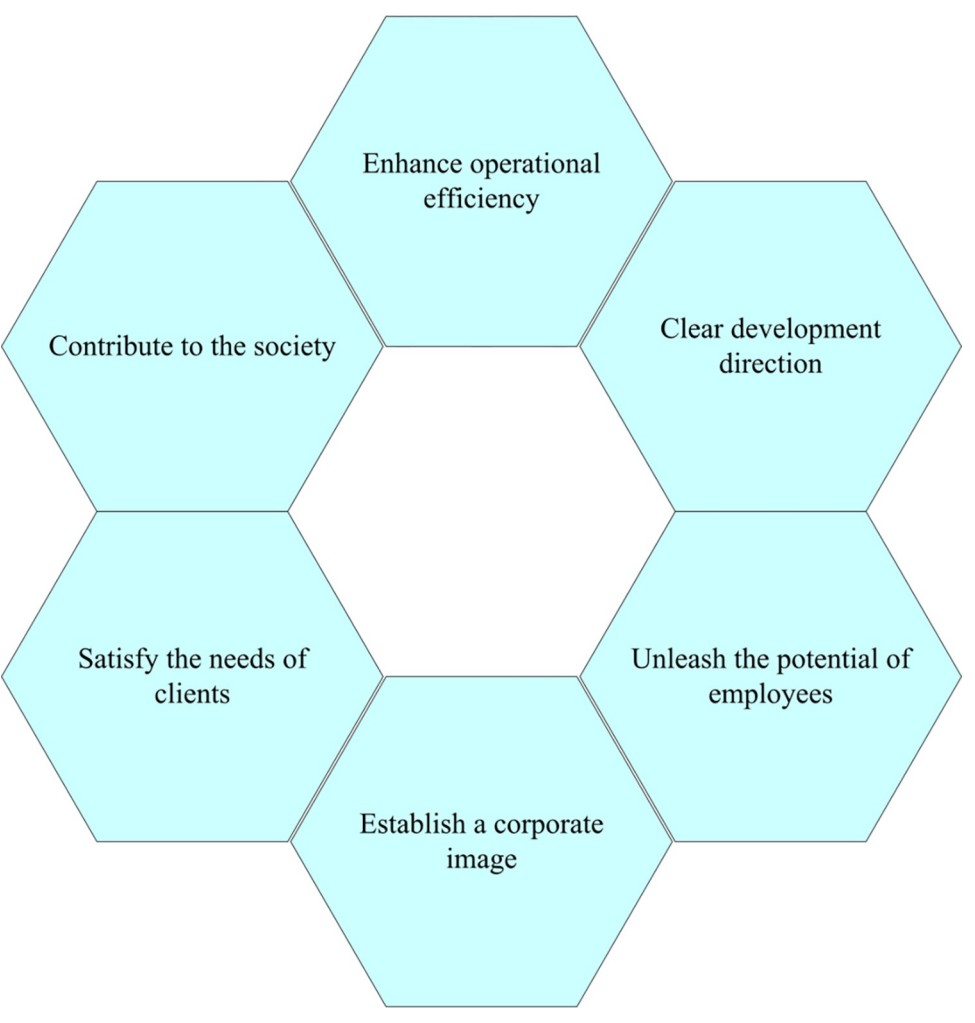

**Fig 2.**

management. In modern enterprise management thinking, enterprises must re-examine their own management methods and models, inherit the advantages and essence of traditional management models, and adapt to the needs of market development in the new era. Enterprise management concepts and models have been updated, business management methods and measures have been optimized, and modern enterprise development needs have been adapted to solve various problems encountered by enterprises [13,14]. By improving the management level of the enterprise, the company can better adapt to the modern management concept, so that the development direction of the company is clearer, the internal operating environment is optimized, and favorable internal conditions are created for the rapid development of the company.

The innovation of the company is promoted.Through continuous innovation and optimized management methods, Apple has promoted the continuous innovation of its products and services and enhanced its market competitiveness. The arrival of the information age creates favorable conditions for enterprises to adapt to the new era [15]. For example, in the process of enterprise management, the management methods of the enterprise should be continuously improved to adapt to the needs of the times and market development, so that the

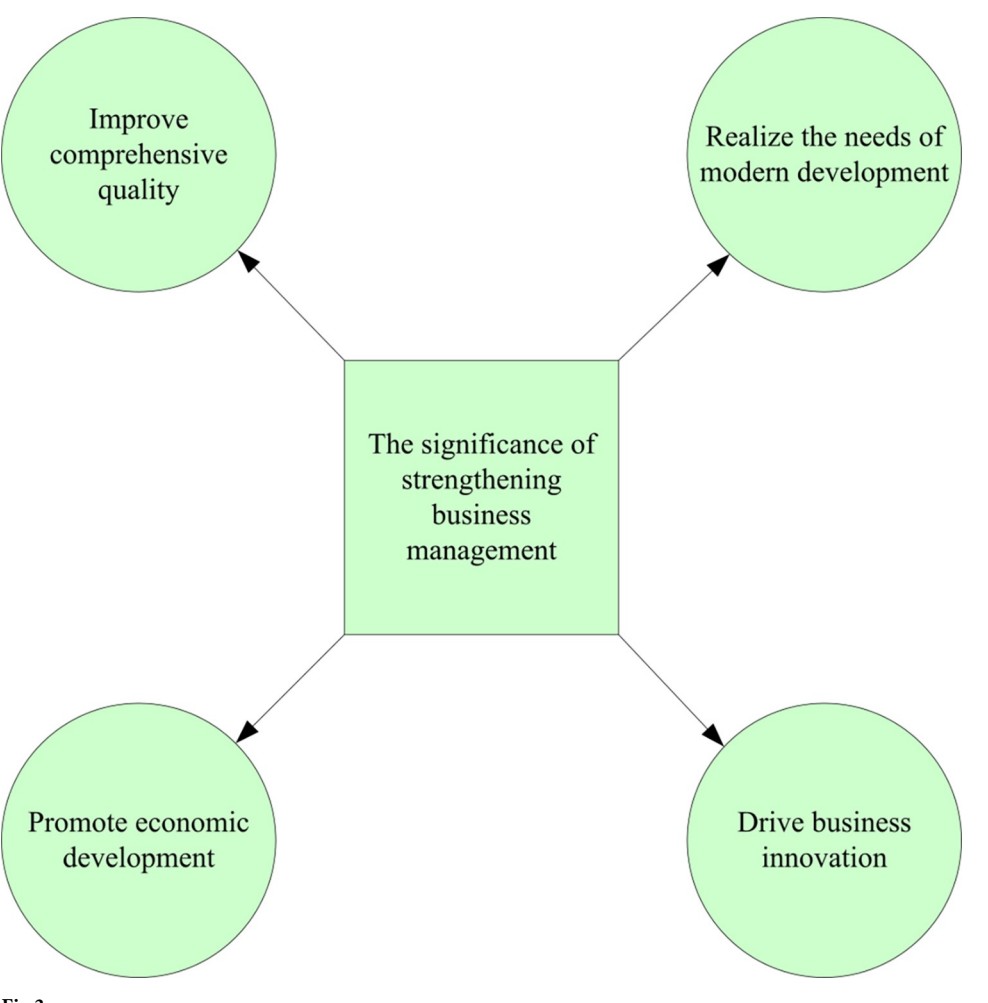

**Fig 3.**

value and role of the business management of the enterprise can be brought into full play, and the basic guarantee for the innovation of enterprise management mode is provided.

The economy is developed.Some studies have shown that effective business management can improve production efficiency, thus driving the economic development of entire industries and countries. Improving the management level of the enterprise can effectively improve the operating efficiency of the enterprise, thereby promoting the development of the company [16]. The realization of the enterprise's economic interests and development goals has played a positive role in promoting the enterprise's position in the market competition and economic development.

The overall quality of the company is improved. The operation and management of an enterprise must be carried out by manpower, and to improve the management ability of an enterprise, it must rely on the management team of the enterprise [17]. The improvement of enterprise management level also reflects the professional quality and ability of enterprise employees. The business management work of the enterprise includes: how to actively and efficiently deal with and resolve risks according to various problems encountered by the enterprise in its business activities; in the process of management, the combination of new management ideas and management methods will change the operation mode of the

enterprise, so that the operation activities of the enterprise can obtain new ideas and blood, thereby stimulating its internal development momentum and potential. Therefore, in order to achieve this goal, it is necessary to improve the overall quality of the entire enterprise to a certain extent, in order to achieve the company's operation and management purposes.Specific improvement objectives include increasing production efficiency, reducing costs, improving product quality, strengthening staff quality, improving customer service levels, and improving environmental and social responsibility. These goals help organizations measure improvements, provide clarity on expected outcomes, and provide direction for management improvement efforts.

Under the new economic situation, by updating the management level of the enterprise and reconstructing the management mode of the enterprise, the management ability of the enterprise can be effectively improved, and the professional skills of the employees can be improved. Therefore, the management and development strategies of the company can be better implemented. The company should continuously improve its internal management system and establish a sound modern management model to enhance the company's management system and make it scientific and systematic [18]. Enterprise managers should improve the operation level of the enterprise as a whole according to the lean management concept to meet the needs of the enterprise's international strategic development, and promote China's reform and opening up. In the fierce market competition, enterprises reduce operating expenses, improve the efficiency of production and operation, thereby enhancing the overall advantages of enterprises.

## Problems in business administration of chinese enterprises

The management system is not perfect. Whether an enterprise can operate effectively depends on the soundness of its management system. The most common situation is that most companies have not been able to fully implement them in the course of their operations. In China, due to the influence of the traditional Chinese family system, there are serious problems in the management system of some family businesses. Due to the lack of close MBA (Master of Business Administration) training organization within the enterprise and the lack of effective communication among personnel, management training is inefficient [19]. According to the Survey Report on the Status Quo of Chinese Enterprise Management (China Academy of Management Science, 2020), about 75% of enterprises believe that their managers lack professional training and development opportunities, which directly affects their management level and performance. In addition, the business management performance appraisal of enterprises also presents a single phenomenon, and the previous performance appraisal system has been unable to meet the needs of the company. If the enterprise only relies on the successful development model and does not consider its own development, it will lead to the inability to innovate the business model of the enterprise, which will lead to the loss of core capabilities of the enterprise, and even major losses. Under the new economic situation, the operation and management system of the enterprise is not perfect, and has a great impact on its operation efficiency. The current common problems in enterprises are shown in Fig 4.

The management ability of enterprise managers is low. Under the new economic situation, the operator's operating ability is directly related to the company's development trend, mental outlook and profitability [20]. However, as far as most of the current enterprises are concerned, due to the organizational structure, the division of functions is not clear. Once a problem occurs, each department will pass the responsibility to others instead of solving the problem. This can only show that the management system formulated by enterprise operators in the course of operation fails to effectively solve various problems in enterprise operation,

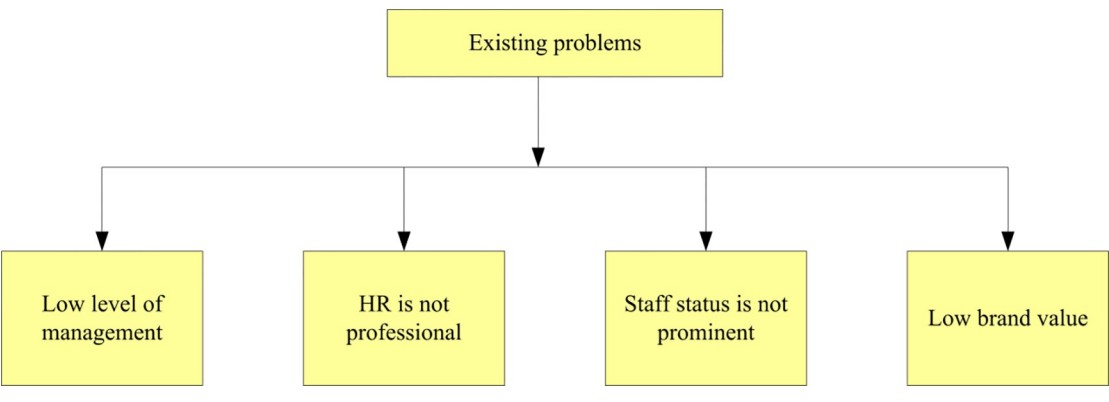

**Fig 4.**

and is also an important reason for the low management ability of managers. Due to the lack of professional management knowledge and technology, it is difficult for enterprises to develop rapidly in today's social environment. Therefore, under the new economic conditions, the management ability of enterprise operators is also a major problem for enterprises. The problems that may be caused by the low level of enterprise management are shown in Fig 5.

The company's human resources lack professionalism. The development of an enterprise is driven by the power of individuals. Therefore, in the selection and employment of talents, the professionalism of the human resources department is particularly important [21]. The human resources department should select personnel who meet the actual needs of the enterprise from the actual situation. It would also be irresponsible to hire not on the basis of personal preference. Under the new economic conditions, the choice of enterprises is not scientific and normative, which is a major problem faced by the operation and management of enterprises.

In an enterprise, the dominant position of employees is not obvious. With the development of society, many excellent companies can see that in the company's management system, it is necessary to pay attention to the economic situation of employees, consider problems from the perspective of employees, prevent leaders from going their own way, and implement humane management. The development of an enterprise requires the joint efforts of everyone. In the

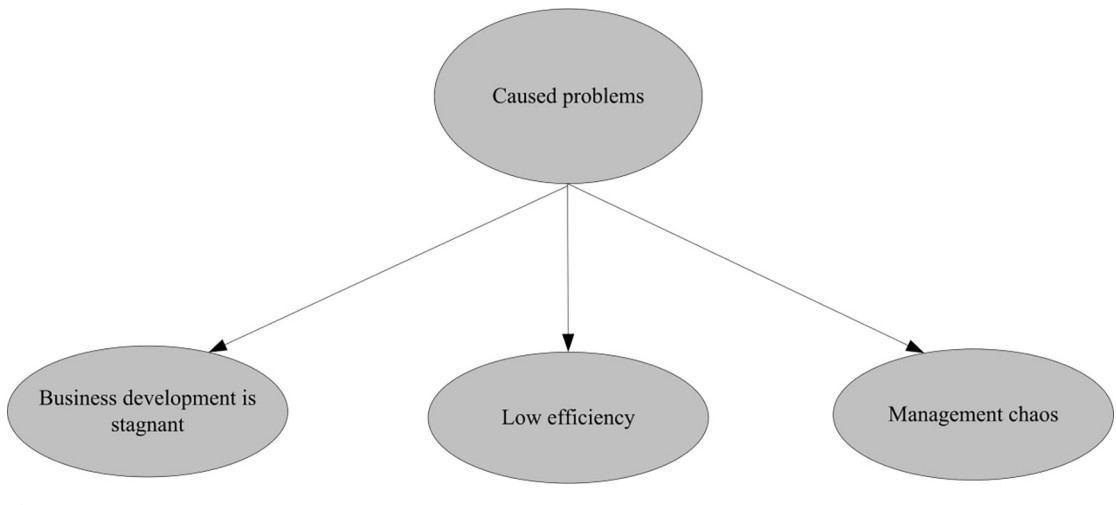

**Fig 5.**

operation of the enterprise, it is necessary to fully reflect the main role of employees, give full play to the role of employees, and mobilize the enthusiasm of employees to make the company more prosperous.

There is no great brand value. Now, China's market economy has undergone great changes. The original restrictive effect on companies has become less and less, and it has become less and less able to play its due role. The regional differences in the product itself will gradually disappear. Therefore, in order to make full use of their own brand advantages, enterprises must innovate their products to realize the rebirth of products and realize their own economic benefits. At present, many companies only focus on the current profit, while ignoring the development of the brand.

## Measures to improve the level of business management of enterprises

In this paper, the measures to improve the business management level of enterprises are mainly explained as follows, as shown in Fig 6.

The concept of business strategy is established. In the new economic environment, enterprises should establish long-term development goals, and gradually improve the modern business management system and internal management rules of the company by optimizing the strategic management system [22]. Enterprises should obtain various historical development indicators from the integration of internal resources, and conduct effective management efficiency analysis on various operating systems to obtain various comparative indicators. The contribution rate of each management link is comprehensively grasped. Enterprises should start from the existing production and operation management mode, reconfigure the operation of the enterprise, fully integrate the managers of various departments, and conduct an all-round investigation on the professional quality, educational background, and skill level of each manager [23,24]. And it is comprehensively inspected to ensure the long-term development of the company and to form a set of effective assessment mechanism. The actual operation and management level of the enterprise is combined with the manager's performance, salary and incentives to establish a post responsibility system, strengthen the incentive effect of the modern enterprise management system on talents, optimize the operation and management of the enterprise, and attach importance to the long-term development of the enterprise [25].

Human resources are optimized. In order to enhance the company's operational management and control capabilities, there must be a company manager team that adapts to the development laws of the modern market economy. The composition of employees of the company's

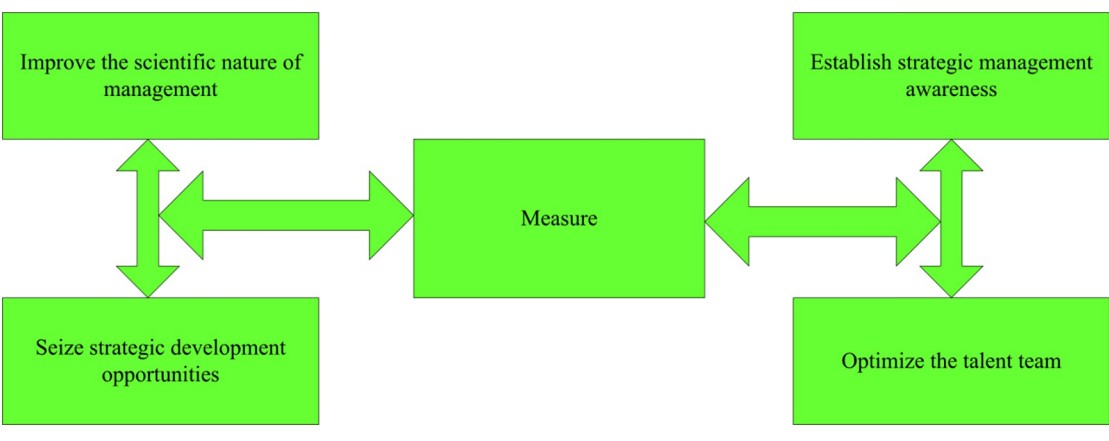

**Fig 6.**

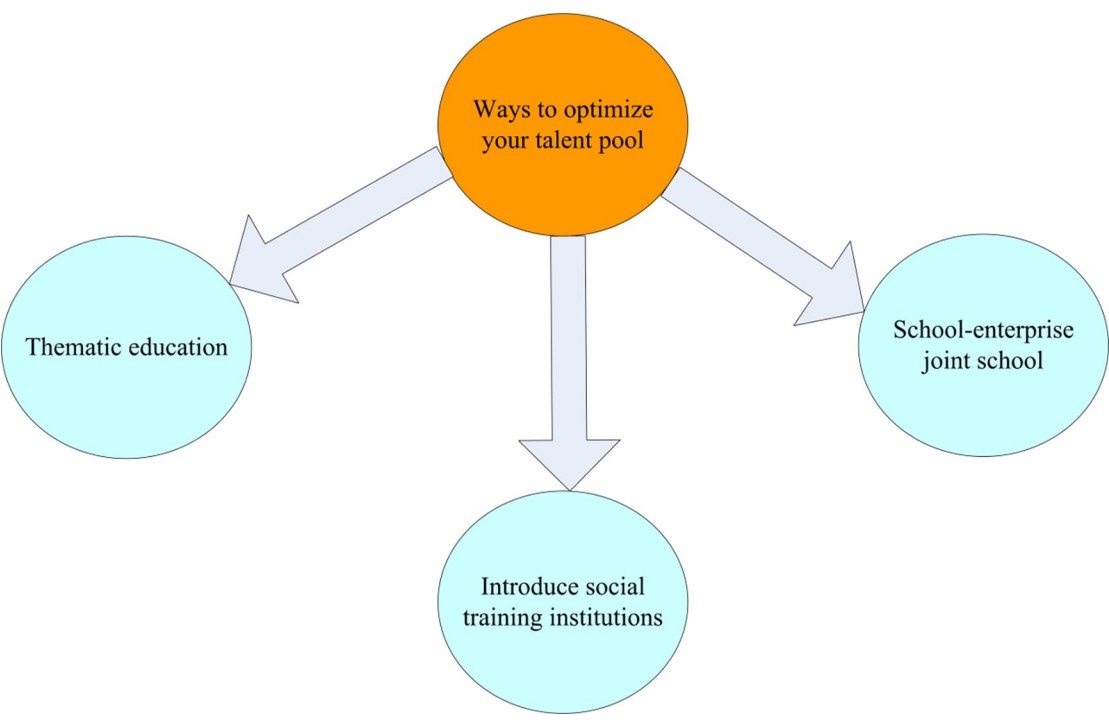

**Fig 7.**

management team is optimized. In the information age, more emphasis is placed on the introduction of new company management with information management power to abandon the negative effect of traditional enterprise management methods on company management, and to adapt to the development trend of the new era. Through the optimization and analysis of the new business environment, the enterprise managers are urged to keep pace with the times, innovate management concepts, and use scientific and effective management methods to systematically analyze the company's business management level. The main methods for optimizing the team structure of professional management talents in enterprises are shown in Fig 7. Specific approaches include strengthening interdisciplinary teamwork, training managers in digital skills, building flexible management systems, driving continuous innovation and iteration, and collaborating with external experts and consultants to ensure that companies maintain a competitive edge and continue to grow in a rapidly changing environment.At the same time, advanced management methods abroad are actively studied. Through special education, intensive discussion, and model training, they can improve the professional quality, skill level and management responsibility of employees. While improving the overall quality of the enterprise in an all-round way, the overall quality and comprehensive quality of the enterprise management team should also be strengthened [26,27].

Changes in national policy are grasped. In the adjustment process of the company's operation and decision-making, it is necessary to make a comprehensive study of China's economic policy, understand the industrial and economic policies of various countries, and the trend of international trade, thereby producing competitive core enterprises. The company must increase the high-tech level of the entire industrial chain according to the strategy of China's industrial development [28,29]. High-tech products are regarded as the core competitiveness of an enterprise, which enhances its technical strength to support the enterprise. The company has always focused on absorbing high-tech talents, further improving the strategic

management ability of the company, and improving the economic benefits of the overall development of the company. In the actual situation of the current enterprise financial control, a new management concept is formed to promote the international management of the enterprise. In order to improve the competitiveness of the company on a global scale, it is necessary to start from the long-term development of the enterprise, and establish advanced management concepts to improve the competitiveness of the enterprise on a global scale. In the increasingly complex situation of international trade, if enterprises want to gain more advantages, they must have more right to speak [30,31].

The level of scientific management has been improved. Enterprises should enhance their understanding of online finance, and use supply chain financing as a means to provide financing for Tuochong companies, optimize resource use, improve overall development capabilities, and establish a modern enterprise management concept. International business methods are actively publicized and promoted, and technical exchanges are carried out with overseas advanced enterprises to improve the level of business management. Enterprises must realize that talents are always the core of the company's development. Only in this way can they attract outstanding talents at home and abroad to participate in the company's development and make their own contributions to the company's development to achieve the best results. Under the new economic situation, enterprises should constantly analyze their own competitiveness, and focus on analyzing problems such as waste of resources, environmental pollution, low efficiency of human resource management, and insufficient cohesion of employees [32]. Under the new economic situation, enterprises should establish a representative and cohesive corporate culture, and condense it into a new function and responsibility through the form of corporate culture to make the enterprise aware of the challenges faced in the new economic environment. In the process of training, they will establish a culture in the hearts of employees, and use compensation and performance management methods to strengthen the company's culture, thereby improving their professional quality.

### Enterprise policy management model

A Nash-balanced corporate hybrid strategy is employed. When there is no pure Nash equilibrium, when both sides of the game have to choose different strategies, the Nash equilibrium problem must be solved. In this paper, a mixed strategy based on Nash equilibrium is proposed to solve the Nash equilibrium problem of enterprises and regulators. The payment matrix for both companies and regulators is:

$$A = \begin{bmatrix} E - D & E - F \\ N & N \end{bmatrix} \tag{1}$$

$$B = \begin{bmatrix} P - F & Q - L \\ -F & -L \end{bmatrix} \tag{2}$$

Among them, E is the profit obtained by the company when it conducts earnings management, and D is the profit caused by the company that does not implement strict control. F is the cost of earnings management by a company that is strictly regulated. N is the profit made by a company that has not implemented earnings management, and P-F is when the company conducts earnings management, and the regulatory authority does not strictly supervise it. Q-L is the effect of the regulator's strict supervision of earnings management on companies. -F is when the company does not have earnings management, and the regulator has imposed less

stringent supervision on it. -L means that when the company does not carry out earnings management, the regulatory authorities impose strict supervision on it.

When a company chooses earnings management, it is best for regulators to take strict supervision; in cases where companies choose not to implement earnings management, the best option for regulators is lax supervision. Represented by Y, the regulators have chosen the possibility of severe regulation, and thus derived their strategic response.

$$y(x) = \begin{cases} 0, x < (L - F)/(Q - P) \\ (0, 1), x = (L - F)/(Q - P) \\ 1, x > (L - F)/(Q - P) \end{cases} \tag{3}$$

It can be seen from this paper that in different intervals, the probability of operating the company is X, and in different intervals, the probability of the regulatory agency implementing strict supervision is Y; when the possibility of the enterprise implementing earnings management is lower than the equilibrium probability, the probability of the regulatory authorities to impose strict supervision on it is 0; when the profitability of the enterprise is H, the probability of the investor accepting the shares is q, and the probability of the enterprise's earnings management is p. It can be concluded that the expected return of the investor accepting the shares is:

$$W = p*q*S(H + R) \tag{4}$$

S is the share and R is the opportunity rate. Then, the expected return for not taking shares is:

$$W_1 = p*S(1 - q)(1 + R) \tag{5}$$

In order to increase management efficiency, enterprises also need to reasonably control their time and expenses. The enterprise management scheduling time is minimized as:

$$\min(\text{cost}) = \sum_i^m C^p(r_i, p_i) + \sum_i^m C^t(\varphi(r_i), \varphi(r_j)) \tag{6}$$

Among them, $C^p(r_i, p_i)$ is the production cost allocated by task $r_i$, and $C^t(\varphi(r_i), \varphi(r_j))$ is the cost of tasks $r_i$ and $r_j$ before and after. The task scheduling of the enterprise production plan needs to satisfy each task before it can be scheduled, that is, the constraint is:

$$\vee D(r_i, p_i) \leq \text{Deadline} \tag{7}$$

$D(r_i, p_i)$ is the minimum time for a business to complete a task. The heuristic algorithm is to generate the scheduling plan according to the minimum cost. The most extreme cost minimum allocation algorithm is when all tasks are allocated to the same enterprise, so that the transportation cost is 0, then the Formula (6) can be:

$$\min(\text{cost}) = \sum_i^m C^p(r_i, p_i) \tag{8}$$

In fact, there is a correlation between the completion time of the enterprise and the cost:

$$\text{cost}(t) = f(\text{time}) \tag{9}$$

The time-based cost minimum scheduling algorithm strategy is: as long as the set of unscheduled tasks is not empty, the task $r_i$ with the smallest $EST(r_i)$ among the unscheduled tasks is selected for scheduling. Through reasonable enterprise management, the efficiency of

the enterprise can be made higher, and it is beneficial to reduce the expenditure and increase the profit of the enterprise.

## Methods of data collection

In the study of enterprise data, questionnaire survey is used in this paper. In order to improve the quality and richness of the results, the following methods can be adopted for research:

Random sampling: The random sampling method is used to ensure the representativeness and statistical reliability of the sample, and to avoid sample bias affecting the results.

Stratified sampling: Respondents are divided into different levels according to specific characteristics (such as industry, size, geographical location, etc.), and then randomly sampled within each level to ensure the diversity and representation of the sample.

Long-term tracking: Not only collect data once, but also conduct long-term tracking surveys to observe the trend of data change over time, so as to better understand the causes and influencing factors of change.

Qualitative research methods: In addition to quantitative data, qualitative data can also be collected, such as open-ended questionnaires, in-depth interviews, etc., to obtain deeper insights and experiences of respondents.

Field survey: While collecting questionnaire data, field observation can also be carried out to observe the behavior and environment of the respondents to supplement the questionnaire survey results.

Cross-validation: Using different data collection methods to verify and supplement results, such as comparing and analyzing questionnaire survey data with experimental data, observation data, etc., to improve the credibility and accuracy of data.

These special research methods can help researchers to understand the research objects more comprehensively, enrich the research results, and improve the scientific and credibility of the research.

## Comparison of management modes in the new economic environment

This paper adopts a rigorous approach to data collection, utilizing structured questionnaires distributed to a diverse range of enterprises within the same region. The objective is to conduct a comprehensive comparison between the business management models prevalent in the new economic environment (referred to as the new model) and the traditional business management models (referred to as the traditional model). Each quarter, 100 companies are surveyed, ensuring a robust dataset spanning three quarters and comprising a total of three experiments. The questionnaires are meticulously designed to capture nuanced insights into various aspects of organizational operations and strategic management practices, ensuring the accuracy and reliability of the collected data. The average of the final results is taken to ensure the accuracy of the data. It mainly compares the complexity of enterprise management process, enterprise management efficiency, management level score, and enterprise quarterly profit [33].

The complexity of business management can also be well illustrated by the two different levels of business administration. The complexity of the enterprise management process affects the efficiency of all aspects of the enterprise, and it also easily leads the enterprise to deviate from the original intention of creating value. From the perspective of enterprise development, enterprises are moving towards refinement, but the time-consuming and complex processes may be counterproductive. The comparison of management process complexity in two different modes is shown in Fig 8. (8A: traditional mode. 8B: new mode).

Fig 8A illustrates that under the traditional mode, the average process complexity between enterprise application and review is 0.7 and 0.66, respectively. Conversely, in Fig 8B, the

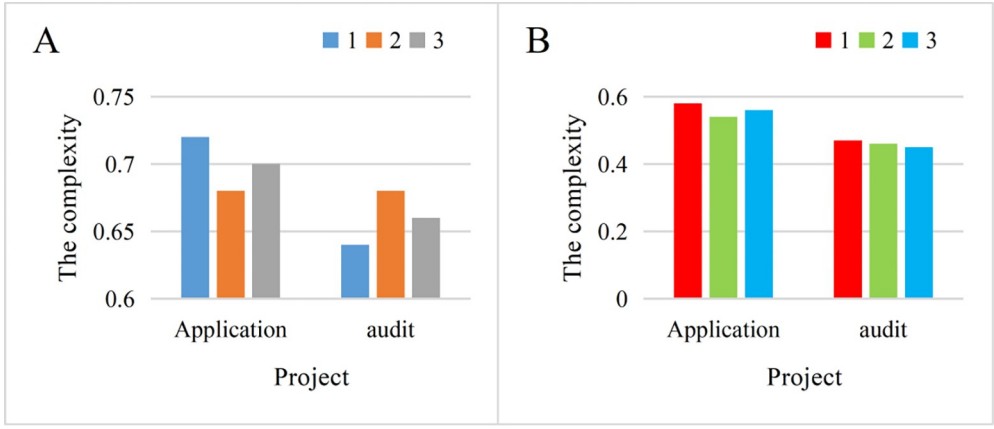

**Fig 8.**

process complexity in the new mode is 0.56 and 0.46, respectively. The higher the complexity, the more intricate the enterprise management process. Compared with the traditional mode, the average complexity has decreased by 0.17. This indicates that business management in the new economic environment tends to have simpler processes, aligning better with current economic trends.Suppose you have two restaurants, one using traditional manual ordering and serving methods, and the other using a new smart ordering system and automated service. In a traditional restaurant, customers fill out a paper menu manually, and the waiter enters the order into the system, which then notifies the kitchen. The process may be subject to misunderstandings, omissions or delays. In contrast, in the new smart restaurant, customers can self-order using a mobile phone app, and the order is sent directly to the kitchen without waiting for a waiter. This intelligent system reduces order errors and processing time, simplifies the entire ordering process, and makes restaurant management more efficient and smooth. This example shows that a new type of intelligent system can reduce the complexity of enterprise management processes and better adapt to the modern economic environment.

The management efficiency of the enterprise will be different under different modes. The management efficiency of the enterprise directly affects the time cost and economic cost of the enterprise, and the management efficiency of the enterprise is also related to the manager and the management object. The comparison of enterprise management efficiency under the two modes is shown in Fig 9. (9A: traditional mode. 9B: new mode).

In Fig 9, the comparison of management efficiency is depicted. In 9A, the three management efficiencies under the traditional mode are 0.57, 0.61, and 0.62, respectively. However, under the new 9B mode, the management efficiency is 0.74, 0.76, and 0.75, respectively. The average management efficiency under the two modes is 0.6 and 0.75, respectively. Compared with the traditional model, the new model has improved management efficiency by 0.15. This highlights that the new model enhances management efficiency for enterprises.Imagine two retail stores, one with traditional manual management and the other with advanced iot technology and data analytics systems to manage inventory and sales. In traditional stores, employees need to manually record inventory and sales data and are prone to errors and underreporting. In contrast, stores that introduce iot technology can monitor inventory levels and sales in real time, and the system automatically predicts demand and replenishes goods in time, avoiding the problem of overstocking or stock shortages. Such an intelligent system greatly improves inventory management and sales efficiency, enabling stores to meet customer needs more quickly and improving overall operating efficiency. This example shows that the

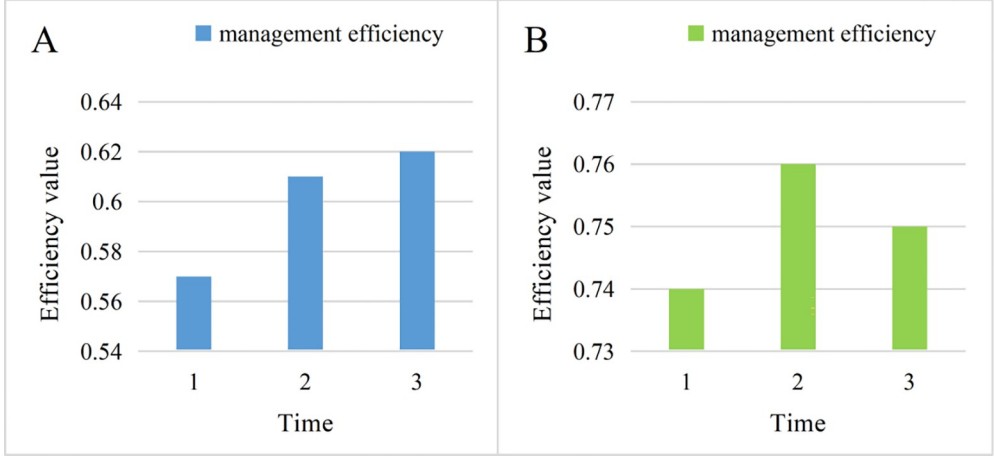

**Fig 9.**

new management model can bring more efficient enterprise management efficiency and enhance the competitiveness of enterprises.

There are also differences in the management level of managers in the two modes. The scores of managers in the enterprise can also illustrate the gap between the two modes. The comparison chart of the management level scores under the two modes is shown in Fig 10. (10A: traditional mode. 10B: new mode).

In Fig 10A, the average scores of the three quarters in the traditional model are 64, 60, and 65 points, respectively. The comprehensive score, calculated as the average of these three scores, is 63 points. Contrarily, under the new 10B model, the average scores for the three quarters are 78, 77, and 76, resulting in an overall score of 77. Compared with the traditional model, the management score has increased by 14 points, indicating that management under the new model better facilitates enterprise operations.

Business management of enterprises is mainly for their better operation, making enterprises develop through decision-making and management, which is also the ultimate goal of enterprises. Fig 11 shows a comparison of the quarterly earnings of companies under the two models.

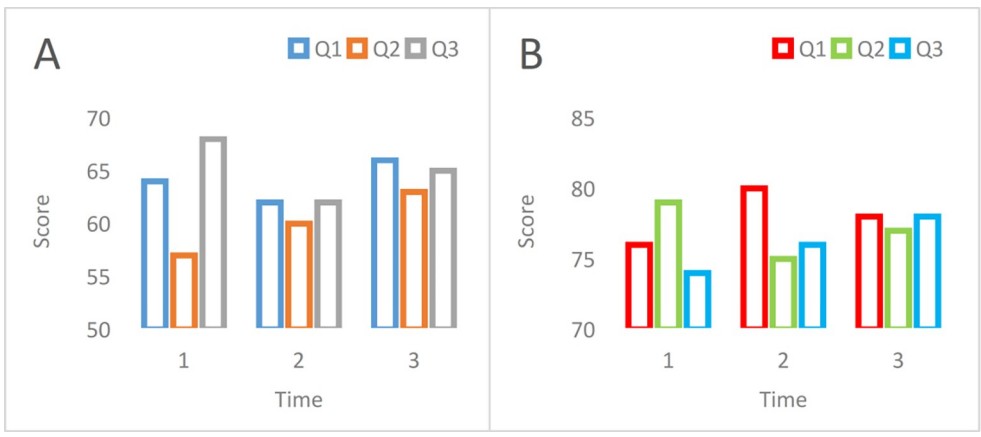

**Fig 10.**

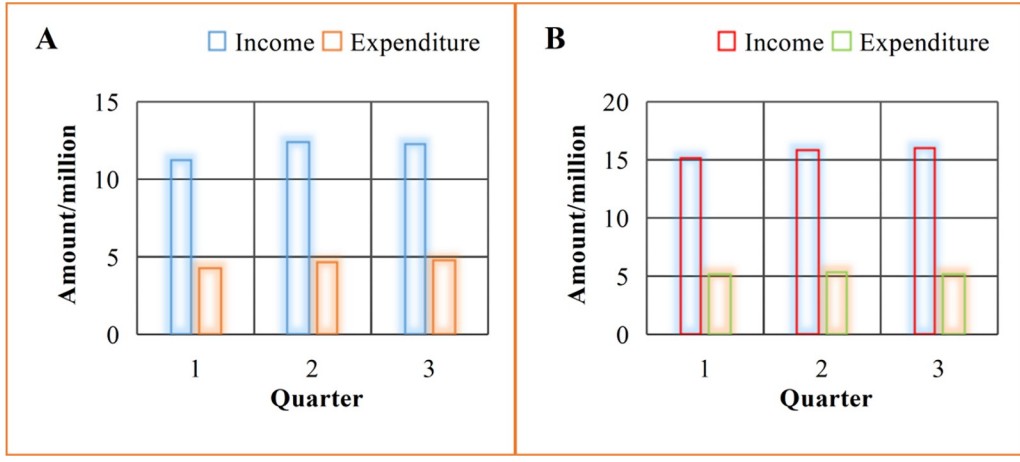

**Fig 11.**

In the traditional model depicted in Fig 11A, the corporate profits for the three quarters are 70,000, 78,000, and 74,000 yuan respectively, with an average quarterly profit of 74,000 yuan. Conversely, in the new model shown in Fig 11B, the corporate profits for the three quarters are 100,000, 104,000, and 108,000 yuan respectively, resulting in an average quarterly profit of 104,000 yuan. Compared with the traditional model, the corporate profit increased by 30,000 yuan. This highlights the enhanced operational efficiency and development potential of the company under the new business model.

## Conclusions

In the face of the new economic environment, the traditional enterprise management model has been unable to meet the current needs. In order to survive and develop in the new environment, we must study the corresponding coping strategies. This paper discusses the current management mode and solution of the enterprise, expounds the importance of improving the enterprise management level, and introduces the enterprise strategic management model for analysis. Through the comprehensive comparison with traditional enterprises in management process complexity, management efficiency, scoring level and quarterly profit, the results show that the enterprise management mode under the new economic environment is more conducive to the survival and development of enterprises, and can better adapt to the challenges of the new environment. The study provides an important reference for business leaders, policy makers, investors and partners, points out the direction of business development in the new economic environment, and enriches the body of knowledge in the field of business management research.Unfortunately, the article does not fully discuss the research of enterprise management strategy under the new economic environment. Due to time constraints, digital transformation, environmental responsibility and globalization strategies could not be explored in depth. In the new economic environment, more companies must reform themselves if they want to survive in order to develop in this era full of risks and opportunities. As the enterprise gradually adapts to the new environment, the business management level of the enterprise will become more comprehensive, and it will also adapt to the more difficult environment.Specific expected improvements, including technology integration and innovation, human resource development, strategic planning and execution, corporate culture and leadership, financial management and risk control, customer relationship management, and

continuous improvement and learning, will help improve the management level of the organization and better respond to challenges and opportunities.

## Supporting information

**S1 File.**
(XLSX)

## Author Contributions

**Writing – original draft:** Zhuolin Xiao.

**Writing – review & editing:** Zhuolin Xiao.

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
