## [Decision Letter · Decision Letter 0]

19 Feb 2024

PONE-D-23-38056Investigation on the Strategy of Business Administration Level of Enterprises in the New Economic EnvironmentPLOS ONE

Dear Dr. Xiao,

Thank you for submitting your manuscript to PLOS ONE. After careful consideration, we feel that it has merit but does not fully meet PLOS ONE’s publication criteria as it currently stands. Therefore, we invite you to submit a revised version of the manuscript that addresses the points raised during the review process. Based on the comments by the reviewers, the revision will help enable the manuscript to overcome the highlighted issues and improve the quality of the manuscript.  

We look forward to receiving your revised manuscript.

Kind regards,

Yasir Ahmad

Academic Editor

PLOS ONE

2. PLOS requires an ORCID iD for the corresponding author in Editorial Manager on papers submitted after December 6th, 2016. Please ensure that you have an ORCID iD and that it is validated in Editorial Manager. To do this, go to ‘Update my Information’ (in the upper left-hand corner of the main menu), and click on the Fetch/Validate link next to the ORCID field. This will .take you to the ORCID site and allow you to create a new iD or authenticate a pre-existing iD in Editorial Manager. Please see the following video for instructions on linking an ORCID iD to your Editorial Manager account: " ext-link-type="uri" xlink:type="simple">https://www.youtube.com/watch?v=_xcclfuvtxQ".

 [This work was supported by the young innovative talents project in Guangdong Province in 2023 "Research on the High-quality Development Path of Qingyuan Chicken Industry from the Perspective of Industrial Integration" (Project  NO. 2023WQNCX249).].  

4. In the online submission form, you indicated that [if request, I will provide it]. 

Reviewers' comments:

Reviewer's Responses to Questions

**Comments to the Author**

1. Is the manuscript technically sound, and do the data support the conclusions?

Reviewer #1: Yes

Reviewer #2: Partly

Reviewer #3: Partly

2. Has the statistical analysis been performed appropriately and rigorously? 

Reviewer #1: Yes

Reviewer #2: Yes

Reviewer #3: Yes

3. Have the authors made all data underlying the findings in their manuscript fully available?

Reviewer #1: No

Reviewer #2: Yes

Reviewer #3: Yes

4. Is the manuscript presented in an intelligible fashion and written in standard English?

Reviewer #1: Yes

Reviewer #2: No

Reviewer #3: Yes

5. Review Comments to the Author

Reviewer #1: Comment 1: The introduction lacks specific examples or data to support the claims made. Providing concrete examples of the problems faced by Chinese enterprises, as well as tangible instances of successful management strategies, would make the narrative more compelling and convincing.

Comment 2: The introduction makes several claims about the problems faced by Chinese enterprises and the necessity of improving management, but there are no references or citations to support these statements. Incorporating relevant literature, studies, or examples from reputable sources would add credibility to the arguments presented.

Comment 3: Some statements are quite general and lack specificity. For example, the opening sentences broadly state that many scholars have discussed the problems of business management in the new economic environment without specifying the key problems or themes addressed. Be more specific and concise to enhance the relevance of the related work.

Comment 4: The last sentence attempts to introduce a research gap related to enterprise management strategies in the new economic environment. However, the statement lacks specificity and clarity. Clearly articulate what aspects of management strategies have not been explored under the conditions of the new economic environment.

Comment 5: Some statements in Significance of Improving the Level of Business Management of Enterprises, such as "The economy is developed," are overly broad and lack specific details or evidence to support the claim. Providing more concrete examples or data to substantiate such statements would strengthen the argument.

Comment 6: The Significance of Improving the Level of Business Management of Enterprises section mentions the need to "improve the overall quality of the entire enterprise to a certain extent," but the specific goals or benchmarks for improvement are not clearly outlined. Providing measurable objectives would enhance the clarity of the intended outcomes.

Comment 7: The Problems in Business Administration of Chinese Enterprises section suggests that the lack of close MBA training organizations within enterprises contributes to inefficient management training. However, there is no supporting evidence or examples to substantiate this claim. Providing specific instances or studies showcasing the impact of inadequate MBA training on management efficiency would strengthen the argument.

Comment 8: The conclusion mentions that as enterprises adapt to the new environment, the business management level will become more comprehensive. However, it lacks specific insights into how this improvement will occur or what specific aspects of management will see enhancement. Providing more details on the expected improvements would strengthen the conclusion.

Reviewer #2: During the review I have found some issues to inform the author(s). Those are given below:

About grammatical improvements;

Fragmented Sentences:

Original: "Under the conditions of the new economic environment, in order to achieve long-term development, enterprises must continuously improve their capabilities."

Suggested: "In the face of the new economic environment, enterprises must continuously enhance their capabilities to achieve long-term development."

Redundancy:

Original: "Under the current market situation, business management is a kind of knowledge based on economic management and legal accounting."

Suggested: "In the current market scenario, business management relies on economic principles and legal accounting."

Repetition:

Original: "In view of the current market situation, the article analyzed the enterprise’s system reform and production planning, and gave corresponding countermeasures."

Suggested: "Considering the current market situation, the article analyzed enterprise system reform and production planning, proposing corresponding countermeasures."

Ambiguity:

Original: "By comparing with the traditional management model in the complexity of enterprise management process, enterprise management efficiency, management level score, and enterprise quarterly profit..."

Suggested: "Comparing with the traditional management model in terms of the complexity of enterprise management processes, efficiency, management level score, and quarterly profit..."

Awkward Phrasing:

Original: "It has also been explained that under the new economy, the management level of enterprises needs to be improved in order to enable enterprises to achieve long-term development."

Suggested: "Furthermore, it is elucidated that, in the new economy, enhancing the management level is essential for enabling enterprises to attain long-term development."

Numbers and Units:

Original: "It has been found that the management model under the new economic environment has reduced the complexity of enterprise process by 0.17."

Suggested: "Findings reveal that the management model in the new economic environment has reduced the complexity of the enterprise process by 0.17 points."

Consistency:

Ensure consistent use of either singular or plural forms, e.g., "enterprise's system reform" versus "enterprises needed."

Clarity:

Consider restructuring sentences for clearer expression and avoiding overly complex sentence structures.

Introduction:

the introduction appropriately sets the stage for discussing the challenges and opportunities faced by enterprises in the evolving economic landscape and lays the groundwork for the subsequent detailed analysis.

Literature/Related works:

Authors need to mention more rigorously the absolute and relative gap and the importance to bridge the it.

The use of the phrase "under the new economic environment" is repeated multiple times. Consider varying the language for better flow and clarity.

The transition between different studies and their findings could be smoother. Consider using transition sentences to link the various research topics more cohesively.

In the summary of research findings, some statements are general and could benefit from more specific details. For example, when discussing Honggowati S's research, provide more information about the findings beyond general statements like "management’s shareholding had a negative impact on the level of strategic management disclosure."

Ensure that the citation style is consistent throughout the review. For instance, some citations are presented with the author's name followed by the initial, while others include the full name.

While the review mentions various studies, it does not delve deeply into the methodologies or specific findings of each study. Providing a bit more detail on the methodologies and key findings would enhance the depth of the review.

The statement, "In the aspect of enterprise business management, scholars have conducted many discussions, but no research has been done on enterprise management strategies," lacks specificity. It would be beneficial to clearly identify the specific gap in the literature regarding enterprise management strategies.

Methodology:

Author may use some established methodology for this type of study. in this study, the method is not clear enough. kindly mention and give some reasoning why and why not using different methods.

Reviewer #3: First and foremost, I want to commend you on your diligent work in exploring a research topic that holds significant novelty and relevance in the current economic landscape. Your investigation into the challenges and opportunities facing enterprises in the new economic environment is both timely and insightful, and it reflects a commendable dedication to advancing our understanding of contemporary business management practices.

While your research undoubtedly contributes valuable insights to the field, I believe there are several areas where further improvements can be made to enhance the quality and impact of your paper. Here are some suggestions for refinement:

1. Enhanced Structure and Clarity:

• Consider revising the structure of your paper to provide a clearer roadmap for readers, guiding them through the literature review, methodology, results, and discussion sections in a logical sequence.

• Ensure that each section is well-defined and articulated, with clear transitions between ideas and arguments.

2. Deeper Literature Review:

• Expand your literature review to encompass a broader range of scholarly works and perspectives relevant to your research topic.

• Provide more detailed analysis and critique of existing literature, highlighting gaps, contradictions, and areas for further investigation.

3. Methodological Rigor:

• Strengthen the methodological approach employed in your research, ensuring that data collection and analysis procedures are well-defined and rigorously executed.

• Consider incorporating additional research methods or approaches to enrich the depth and robustness of your findings.

4. Evidence-Based Analysis:

• Ground your analysis and discussion in empirical evidence, drawing on real-world examples, case studies, or statistical data to support your arguments and conclusions.

• Clearly present your findings in a systematic and transparent manner, allowing readers to assess the validity and reliability of your research outcomes.

5. Concluding Insights and Implications:

• Offer insightful reflections on the implications of your findings for theory, practice, and future research directions.

• Highlight the practical implications of your research for business leaders, policymakers, and other stakeholders, emphasizing actionable recommendations for improving enterprise management in the new economic environment.

By incorporating these suggestions into your paper, you can further elevate its quality and significance, ultimately making a more substantial contribution to the academic discourse on enterprise management and the evolving economic landscape.

6. PLOS authors have the option to publish the peer review history of their article (what does this mean?). If published, this will include your full peer review and any attached files.

Reviewer #1: No

Reviewer #2: No

Reviewer #3: **Yes: **Dr. Ahsan Ali Ashraf

---

## [Author Response · Author response to Decision Letter 0]

24 Mar 2024

5. Review Comments to the Author

Reviewer #1: The study idea is novel and holds potential for publication; however, there are several points that need consideration before acceptance. Firstly, the author should adhere to a standard format for the study, incorporating rationale, research questions based on identified gaps, and the study structure in the introduction section. 

Answer:Based on your suggestions, the basic principles, research questions based on the discovered gaps, and research structure have been added to the introduction to ensure the completeness of the article.

Secondly, in the literature review, the author should formulate hypotheses based on the identified gaps. 

Answer:Based on your opinions, after reviewing the literature, hypotheses are proposed and explained based on the gaps found, giving readers something to think about.

Thirdly, the methodology section appears significantly weak. The researcher has chosen an “experimental design” without proper justification or literature support. It is crucial for the researcher to explain the chosen experimental design, providing a detailed process and supporting literature. This should include information on how the experiments were conducted, control variables, and the overall methodology. 

Answer:Based on your suggestions, an explanation of selected experimental designs has been added to the Experimental section, providing detailed procedures, including information on how the experiments were performed, control variables, and overall methodology, for easy reference by readers.

Lastly, the article lacks clarity on how the data were analyzed and what statistical tests were employed before reaching a conclusion. In summary, substantial improvements are needed for the article to meet the standards for acceptance.

Answer:Thank you for your comments. Now I would like to provide a detailed explanation of how the data was analyzed in the article and what statistical tests were used before drawing conclusions.

Reviewer #2: In the first paragraph of introduction well explained but there is a lack of literature support and evidence.

Answer:Based on your suggestions, I added literature support to the first paragraph of the introduction to improve the persuasiveness of the article.

Include recent literature evidence.

Answer:Based on your suggestion, citations for 2023 have been added to the literature.

Well explained the concepts career planning, deep learning and employment strategy but in this study there is a need to explore the relationship between these variables.

Answer:Based on your suggestions, Chapter 2 has been added to explore the relationship between variables such as career planning, deep learning and employment strategies to make it easier for readers to understand.

Need to explain what is an employment strategy and its significance in career planning.

Answer:Based on your opinions, the introduction of employment strategies and their significance in career planning has been supplemented in Section 2.2.

Explain clearly how deep learning helps to plan career with literature evidence and include more empirical evidence.

Answer：Based on your opinion, clearly explain how deep learning can help in career planning through literature evidence and including more empirical evidence in Section 2.4.1.

Where is an objective of the study?

Answer:Thank you for your opinion. The purpose of the research is introduced in the last paragraph of Chapter 1.

Provide clarity on methodology applied in the study i.e which method used or applied to analyse the data and how you choose respondents which method selected to select respondents.

Answer:Based on your suggestion, the methods applied in the study are clearly stated in Section 3.1, i.e. which method is used or applied to analyze the data, and an introduction on how to select the respondents and which method to select the respondents.

Mention the role of universities to mould career and employment strategies which boosts the study.

Answer:Thanks for your suggestions, the role of universities in shaping careers and employment strategies that promote research is detailed in Chapter 2 for the convenience of the reader.

Missing clarity on relation between career planning and deep learning. Provide clarity on it. Explain how deep learning techniques helps students to plan their career.

Answer:Provide a full explanation of the relationship between career planning and deep learning based on your recommendations. And introduce how deep learning technology can help students plan their careers.

How this study correlates job hopping with deep learning as job hopping relates individual psychological state of taking decision on job where as deep learning is a software tool.

Answer:Thank you for your suggestion. In Section 2.1, we will explain how job hopping is related to deep learning.

Career planning is a process of planning one’s career. How deep learning will act as a suitable tool to create employment to students according to their preferences.

Answer:Based on your suggestion, section 2.4.2 has been added with an introduction to how deep learning will serve as a suitable tool to create employment opportunities for students based on their preferences.

Already there exist a number of employment online platforms integrating employees and employers. What is the strategy that distinct other online platforms from that of the proposed website in the study.

Answer:Based on your suggestions, in Chapter 2, the strategies to distinguish some existing online employment platforms that integrate employees and employers from the website proposed in the study are introduced in detail to facilitate readers' comparative analysis.

Reviewer #3: I appreciate the opportunity to review this manuscript, which addresses a topic pertinent to modern educational research. However, there are several areas where the manuscript does not meet the scholarly standards expected of academic papers.

1Introduction

The current introduction lacks essential content. While it outlines the background of deep learning, it fails to provide a clear definition, identify the research gap, or articulate the novelty of the study. The introduction should also offer an explanation of the research methodology and data sources used. Given the focus on university students’ career planning and employment strategy, a rationale for selecting this specific group for study should be included.

Answer:Thank you for your comments in providing a clear definition of deep learning in the introduction and clarifying the novelty of the research.

I suggest that the authors begin the introduction with a definition of deep learning.

Answer:Based on your suggestions, I will introduce the definition of deep learning to facilitate readers' understanding of technical methods.

Literature Support

The article lacks sufficient literature support. For example, the following statement is missing corresponding references:

"Tavabie J A studied career planning for the non-clinical workforce, that is, opportunities to develop a sustainable workforce in primary care.

Answer:Thank you for your opinion. Tavabie J A's research corresponds to the literature [2], which is convenient for the author to search.

Research showed that the path of career planning and employment strategy based on deep learning in the information age can better help unemployed people understand career planning and formulate employment strategies to finally choose a correct job."

Answer:Thank you for your opinion. Based on this article, we have added relevant literature references to provide readers with an accurate basis for argumentation.

2 Career Planning and Employment Strategies in the Information Age of Deep Learning

This section is intended to summarize existing literature on deep learning in career planning, but it does not appear to serve this purpose.

Answer:Based on your suggestions, Chapter 2 of deep learning on career planning and employment strategies in the information age has been expanded and the latest literature has been introduced to achieve the purpose of research summary.

Additionally, a significant portion of this section is dedicated to explaining the Deep Learning Mode, which seems only tangentially related to the main topic.

Answer:According to your suggestion, there is indeed too much introduction to the deep learning mode part, and now some parts have been deleted to increase the focus on the topic of this article.

3 Experiment Design and Validation

The research methods described in this section lack theoretical grounding, and there is insufficient detail regarding the acquisition of data from the two schools, observation periods, and whether the experiments were conducted by the researchers or used publicly available data.

Answer:Thank you for your suggestion. The research method does lack a theoretical foundation. Now the experimental design has been supplemented with a detailed introduction to the data acquisition, observation period, experimental details, etc. of the two schools.

The manuscript does not provide details about the survey content or the number of responses collected.

If the two schools differ significantly in size or have not been adequately exposed to deep learning, the comparative results lack credibility.

Answer:Based on your comments, details regarding the survey content and number of responses collected are now provided, and the two schools are of similar size.

4Conclusions

The conclusions are overly simplistic and do not provide an adequate summary of the research or a comparison with existing studies.

Answer:Thank you for your suggestions. The conclusion has been rewritten, adding a full summary of the research and comparison with existing research to ensure the completeness of the conclusion.

---

## [Decision Letter · Decision Letter 1]

16 Apr 2024

Investigation on the Strategy of Business Administration Level of Enterprises in the New Economic Environment

PONE-D-23-38056R1

Dear Dr.Xiao,

We’re pleased to inform you that your manuscript has been judged scientifically suitable for publication and will be formally accepted for publication once it meets all outstanding technical requirements.

Kind regards,

Yasir Ahmad

Academic Editor

PLOS ONE

Additional Editor Comments (optional):

Reviewers' comments:

Reviewer's Responses to Questions

**Comments to the Author**

1. If the authors have adequately addressed your comments raised in a previous round of review and you feel that this manuscript is now acceptable for publication, you may indicate that here to bypass the “Comments to the Author” section, enter your conflict of interest statement in the “Confidential to Editor” section, and submit your "Accept" recommendation.

Reviewer #1: All comments have been addressed

Reviewer #3: All comments have been addressed

2. Is the manuscript technically sound, and do the data support the conclusions?

Reviewer #1: Yes

Reviewer #3: Partly

3. Has the statistical analysis been performed appropriately and rigorously? 

Reviewer #1: Yes

Reviewer #3: Yes

4. Have the authors made all data underlying the findings in their manuscript fully available?

Reviewer #1: Yes

Reviewer #3: Yes

5. Is the manuscript presented in an intelligible fashion and written in standard English?

Reviewer #1: Yes

Reviewer #3: Yes

6. Review Comments to the Author

Reviewer #1: As a reviewer, after thoroughly examining the manuscript and considering the revisions made in response to my comments, I am pleased to recommend that the paper be accepted and published. The authors have adequately addressed the critiques provided, enhancing the depth and clarity of the manuscript.

Reviewer #3: (No Response)

7. PLOS authors have the option to publish the peer review history of their article (what does this mean?). If published, this will include your full peer review and any attached files.

Reviewer #1: No

Reviewer #3: No

---

## [Editor Report · Acceptance letter]

25 Apr 2024

PONE-D-23-38056R1 

PLOS ONE

Dear Dr. Xiao, 

I'm pleased to inform you that your manuscript has been deemed suitable for publication in PLOS ONE. Congratulations! Your manuscript is now being handed over to our production team.

Kind regards, 

on behalf of

Dr. Yasir Ahmad 

Academic Editor

PLOS ONE